# Speeding up Training of Linear Predictors for Multi-Antenna Frequency-Selective Channels via Meta-Learning

**DOI:** 10.3390/e24101363

**Published:** 2022-09-26

**Authors:** Sangwoo Park, Osvaldo Simeone

**Affiliations:** Department of Engineering, King’s College London, London WC2R 2LS, UK

**Keywords:** channel prediction, meta-learning, multi-antenna frequency-selectivity, equilibrium propagation

## Abstract

An efficient data-driven prediction strategy for multi-antenna frequency-selective channels must operate based on a small number of pilot symbols. This paper proposes novel channel-prediction algorithms that address this goal by integrating transfer and meta-learning with a reduced-rank parametrization of the channel. The proposed methods optimize linear predictors by utilizing data from previous frames, which are generally characterized by distinct propagation characteristics, in order to enable fast training on the time slots of the current frame. The proposed predictors rely on a novel long short-term decomposition (LSTD) of the linear prediction model that leverages the disaggregation of the channel into long-term space-time signatures and fading amplitudes. We first develop predictors for single-antenna frequency-flat channels based on transfer/meta-learned quadratic regularization. Then, we introduce transfer and meta-learning algorithms for LSTD-based prediction models that build on equilibrium propagation (EP) and alternating least squares (ALS). Numerical results under the 3GPP 5G standard channel model demonstrate the impact of transfer and meta-learning on reducing the number of pilots for channel prediction, as well as the merits of the proposed LSTD parametrization.

## 1. Introduction

The capacity to accurately predict channel state information (CSI) is a key enabler of proactive resource allocation strategies, which are central to many visions for efficient and low-latency communications in 6G and beyond (see, e.g., [1]). The problem of channel prediction is relatively straightforward in the presence of known channel statistics. In fact, under the common assumption that multi-antenna frequency-selective channels follow stationary complex Gaussian processes, optimal channel predictors can be obtained via linear minimum mean squared error (LMMSE) estimators such as the Wiener filter [2]. However, in practice, the channel statistics are not known, and predictors need to be optimized based on training data obtained through the transmission of pilot signals [3,4,5,6,7,8]. The problem addressed by this paper concerns the design of data-efficient channel predictors for multi-antenna frequency-selective channels.

### 1.1. Context and Prior Art

A classical approach to tackle this problem is to optimize finite impulse response (FIR) filters [9], or recursive linear filters via autoregressive (AR) models [3,4,5] such as Kalman filtering (KF) [6,7,8], by estimating channel statistics from the available pilot data. Although recursive linear filters generally outperform FIR filters when an accurate model of the state-transition dynamics is available [10,11], FIR filters are typically advantageous in the presence of limited amounts of pilot data [12]. More recently, deep learning-based nonlinear predictors have also been proposed to adapt to channel statistics through the training of neural networks, namely recurrent neural networks [13,14,15,16], convolutional neural networks [17,18], and multi-layer perceptrons [19].

As reported in [14,15,16,19], deep learning-based predictors tend to require larger training (pilot) data, and fail to outperform well-designed linear filters in the low-data regime. Some solutions addressing this issues include [20], which applies reinforcement learning to determine whether to predict channels or not at the current time, and the use of hypernetworks to adapt parameters of a KF accordingly to current channel dynamics [12].

Most prior work, with the notable exception of [12], focuses on the optimization of channel predictors under the assumption of a stationary spatio-temporal correlation function across the time interval of interest. This conventional approach fails to leverage common structure that may exist across multiple frames, with each frame being characterized by distinct spatio-temporal correlations (see Figure 1). Reference [12] allowed for varying Doppler spectra across frames, through a deep learning-based hypernetwork that is used to adapt the parameters of a generative model [12].

This paper takes a different approach that allows us to move beyond the single-antenna setting studied in [12]. As described in the next subsection, key ingredients of the proposed methods are transfer learning and meta-learning. Transfer learning [21] and meta-learning [22] aim at using knowledge from distinct tasks in order to reduce the data requirements on a new task of interest. Given a large enough resemblance between different tasks, both transfer learning and meta-learning have shown remarkable performance to reduce the sample complexity in general machine learning problems [23]. Transfer learning applies to a specific target task, whereas meta-learning caters to adaptation to any new task (see e.g., [24]).

Previous applications of transfer learning to communication systems include beamforming for multi-user, multiple-input, single-output (MISO) downlink [25] and for intelligent reflecting surfaces (IRS)-assisted MISO downlink [26], and downlink channel prediction [27,28] (see also [25,27]). Meta-learning has been applied to communication systems, including demodulation [29,30,31,32], decoding [33], end-to-end design of encoding and decoding with and without a channel model [34,35]; MIMO detection [36], beamforming for multiuser MISO downlink systems via [37], layered division multiplexing for ultra-reliable communications [38], UAV trajectory design [39], and resource allocation [40].

### 1.2. Contributions

This paper proposes novel efficient data-driven channel prediction algorithms that reduce pilot requirements by integrating transfer and meta-learning with a novel long-short-term decomposition (LSTD) of the linear predictors. Unlike the prior articles reviewed above, the proposed methods apply to multi-antenna frequency-selective channels whose statistics change across frames (see Figure 1). Specific contributions are as follows.

We develop efficient predictors for single-antenna frequency-flat channels based on transfer/meta-learned quadratic regularization. Transfer and meta-learning are used to leverage data from multiple frames in order to extract shared useful knowledge that can be used for prediction on the current frame (see Figure 2).Targeting multi-antenna frequency-selective channels, we introduce the LSTD-based model class of linear predictors that builds on the well-known disaggregation of standard channel models into long-term space-time signatures and fading amplitudes [5,41,42,43,44]. Accordingly, the channel is described by multipath features, such as angle of arrivals, delays, and path loss, that change slowly across the frame, as well as by fast-varying fading amplitudes. Transfer learning and meta-learning algorithms for LSTD-based prediction models are proposed that build on equilibrium propagation (EP) and alternating least squares (ALS).Numerical results under the 3GPP 5G standard channel model demonstrate the impact of transfer and meta-learning on reducing the number of pilots for channel prediction, as well as the merits of the proposed LSTD parametrization.

Part of this paper was presented in [45], which only covered meta-learning for the case of single-antenna frequency-flat channels. As compared to [45], this journal version includes both transfer and meta-learning, and it addresses the general scenario of multi-antenna frequency-selective channels by introducing and developing the LSTD model class of linear predictors.

### 1.3. Organization

The rest of the paper is organized as follows. In Section 2, we detail system and channel models, and describe conventional, transfer, and meta-learning concepts. In Section 3, we develop solutions for single-antenna frequency-flat channels. In Section 4, multi-antenna frequency-selective channels are considered, and we propose LSTD-based linear prediction schemes. Numerical results are presented in Section 5, and conclusions are presented in Section 6.

Notation: In this paper, ${\left(\cdot \right)}^{⊤}$ denotes the transposition; ${\left(\cdot \right)}^{†}$ the Hermitian transposition, ${\left(\cdot \right)}_{F}$ the Frobenius norm, |·| the absolute value, ||·|| the Euclidean norm, vec(·) the vectorization operator that stacks the columns of a matrix into a column vector, ${\left[\cdot \right]}_{i}$ the *i*-th element of the vector, and $I_{S}$ the S×S identity matrix for some integer *S*.

## 2. System Model

### 2.1. System Model

As shown in Figure 1, we study a frame-based transmission system, with each frame containing multiple time slots. Each frame carries data from a possibly different user to the same receiver, e.g., a base station. The receiver has $N_{R}$ antennas, and the transmitters have $N_{T}$ antennas. The channel $h_{l,f}$ in slot l=1,2,… of frame f=1,2,… is a vector with $S=N_{R}N_{T}W$ entries, with *W* being the delay spread measured in number of transmission symbols within each frame *f*, the multi-path channels $h_{l,f}\in C^{N_{R}N_{T}W\times 1}$ are characterized by fixed, frame-dependent, average path powers, path delays, Doppler spectra, and angles of arrival and departure [46]. For instance, in a frame *f*, we may have a slow-moving user in line-of-sight condition subject to time-invariant fading, whereas in another, the channel may have significant scattering with fast temporal variations with a large Doppler frequency. In both cases, the frame is assumed to be short enough that average path powers, path delays, Doppler spectra, and angles of arrival and departure do not change within the frame [41,42].

As also seen in Figure 1, for each frame *f*, we are interested in addressing the lag-δ channel prediction problem, in which channel $h_{l+\delta ,f}$ is predicted based on the *N* past channels:(1)$$\begin{array}{c} H_{l,f}^{N}=\left[h_{l,f},\ldots ,h_{l-N+1,f}\right]\in C^{S\times N}. \end{array}$$ We adopt linear prediction with regressor $V_{f}\in C^{SN\times S}$, so that the prediction is given as
(2)$$\begin{array}{c} {\hat{h}}_{l+\delta ,f}=V_{f}^{†}\mathrm{vec}\left(H_{l,f}^{N}\right). \end{array}$$ The focus on linear prediction is justified by the optimality of linear estimation for Gaussian stationary processes [47], which provide standard models for fading channels in rich scattering environments.

Assuming no prior knowledge of the channel model, we adopt a data-driven approach to the design of the predictor (Equation 2). Accordingly, to train the linear predictor (Equation 2), for any frame *f*, the receiver is assumed to have available the training set
(3)$$\begin{array}{c} Z_{f}^{\mathrm{tr}}={\left\{\left(x_{i,f},y_{i,f}\right)\right\}}_{i=1}^{L^{\mathrm{tr}}}\equiv {\left\{\left(\mathrm{vec}\left(H_{l,f}^{N}\right),h_{l+\delta ,f}\right)\right\}}_{l=N}^{L^{\mathrm{tr}}+N-1} \end{array}$$
encompassing $L^{\mathrm{tr}}$ input–output examples. Dataset $Z_{f}^{\mathrm{tr}}$ can be constructed from $L^{\mathrm{tr}}+N+\delta -1$ channels $\left\{h_{1,f},\ldots ,h_{L^{\mathrm{tr}}+N+\delta -1,f}\right\}$ by using the lag-δ channel $h_{l+\delta ,f}$ as label for the covariate vector $\mathrm{vec}(H_{l,f}^{N})$. In practice, the channel vectors $h_{l,f}$ are estimated by using pilot symbols, and estimation noise can be easily incorporated in the model (see Section 2.5). Throughout, we implicitly assume that the channels $h_{l,f}$ correspond to estimates available at the receiver.

From dataset $Z_{f}^{\mathrm{tr}}$ in (Equation 3), we write the corresponding $L^{\mathrm{tr}}\times SN$ input matrix $X_{f}^{\mathrm{tr}}={\left[x_{1,f}^{†},\ldots ,x_{L^{\mathrm{tr}},f}^{†}\right]}^{⊤}$, and the $L^{\mathrm{tr}}\times S$ target matrix $Y_{f}^{\mathrm{tr}}={\left[y_{1,f}^{†},\ldots ,y_{L^{\mathrm{tr}},f}^{†}\right]}^{⊤}$, so that the dataset can be expressed as the pair $Z_{f}^{\mathrm{tr}}=\left(X_{f}^{\mathrm{tr}},Y_{f}^{\mathrm{tr}}\right)$.

### 2.2. Channel Model

We adopt the standard spatial channel model [46]. Accordingly, a channel vector $h_{l,f}$ for slot *l* in frame *f*, is obtained by sampling the continuous-time multipath vector channel impulse response
(4)$$\begin{array}{c} h_{l,f}\left(\tau \right)=\sum_{d=1}^{D}\sqrt{\Omega_{d,f}}a_{d,f}g\left(\tau -\tau_{d,f}\right)\exp\left(-j2\pi \gamma_{d,f}t_{l}\right), \end{array}$$
which is the sum of contributions from *D* paths. In (Equation 4), the waveform g(τ) is given by the convolution of the transmitted waveform and the matched filter at the receiver. Furthermore, the contribution of the *d*-th path depends on the average power $\Omega_{d,f}$, the path delay $\tau_{d,f}$, the $N_{T}N_{R}\times 1$ spatial vector $a_{d,f}$, the Doppler frequency $\gamma_{d,f}$, and the starting wall-clock time of the *l*-th slot $t_{l}$. The average power $\Omega_{d,f}$, path delays $\tau_{d,f}$, spatial vector $a_{d,f}$, and Doppler frequency $\gamma_{d,f}$ are constant within one frame because they depend on large-scale geometric features of the propagation environment. However, they may change over frames following Clause 7.6.3.2 (Procedure B) in [46]. The number of paths is assumed without loss of generalization to be the same for all frames *f* because one can set $\Omega_{d,f}=0$ for frames with a smaller number of paths.

In [46], the spatial vector $a_{d,f}$ has a structure that depends on field patterns and steering vectors of the transmit and receive antennas, as well on the polarization of the antennas. Mathematically, the entry of the spatial vector $a_{d,f}$ corresponding to the receive and transmit antenna element $n_{R}$ and $n_{T}$ can be modeled as [46]
(5)$$\begin{array}{c} {\left[a_{d,f}\right]}_{n_{R}+\left(n_{T}-1\right)N_{R}}=F_{rx,n_{R}}{\left(\theta_{d,f,ZOA},\phi_{d,f,AOA}\right)}^{T}M_{d,f}\\ \cdot F_{tx,n_{T}}\left(\theta_{d,f,ZOD},\phi_{d,f,AOD}\right)\exp\left(-\frac{j2\pi l_{d,f,n_{R},n_{T}}}{\lambda_{0}}\right), \end{array}$$
where $F_{rx,n_{R}}\left(\cdot ,\cdot \right)$ and $F_{tx,n_{T}}\left(\cdot ,\cdot \right)$ are the 2×1 field patterns, $\theta_{d,f,ZOA}$, $\phi_{d,f,AOA}$, $\theta_{d,f,ZOD}$, and $\phi_{d,f,AOD}$ are the zenith angle of arrival (ZOA), azimuth angle of arrival (AOA), zenith angle of departure (ZOD), and azimuth angle of departure (AOD) (in degrees), $\lambda_{0}$ is the wavelength (in m) of the carrier frequency, $l_{d,f,n_{R},n_{T}}$ is the length of the path (in m) between the two antennas, and $M_{d,f}$ is the polarization coupling matrix defined as
(6)$$\begin{array}{c} M_{d,f}=\left(\begin{array}{c} \;\;\exp\left(j\Phi_{d,f}^{\theta \theta }\right)\;\;\sqrt{1/\kappa_{d,f}}\exp\left(j\Phi_{d,f}^{\theta \phi }\right)\\ \sqrt{1/\kappa_{d,f}}\exp\left(j\Phi_{d,f}^{\phi \theta }\right)\;\;\exp\left(j\Phi_{d,f}^{\phi \phi }\right) \end{array}\right), \end{array}$$
with random initial phase $\Phi_{d,f}^{\left(\cdot ,\cdot \right)}∼U\left(-\pi ,\pi \right)$ and log-normal distributed cross polarization power ratio (XPR) $\kappa_{d,f}>0$ [46].

In order to obtain the S×1 vector $h_{l,f}$, we sample the continuous-time channel $h_{l,f}\left(\tau \right)$ in (Equation 4) at Nyquist rate 1/T to obtain *W* discrete-time $N_{R}N_{T}\times 1$ channel impulse response
(7)$$\begin{array}{c} h_{l,f}\left[w\right]=h_{l,f}\left(\left(w-1\right)T\right) \end{array}$$
for w=1,…,W. Following [41], the channel vector $h_{l,f}\in C^{N_{R}N_{T}W\times 1}$ is obtained by concatenating the *W* channel vectors $h_{l,f}\left[w\right]$ for w=1,…,W as
(8)$$\begin{array}{c} h_{l,f}={\left[h_{l,f}\left[1\right],\ldots ,h_{l,f}\left[W\right]\right]}^{⊤}. \end{array}$$

### 2.3. Conventional Learning

The optimization of the linear predictor $V_{f}$ in (Equation 2) can be formulated as a supervised learning problem as it will be detailed in Section 3. In conventional learning, the predictor $V_{f}$ is designed separately in each frame *f* based on the corresponding dataset $Z_{f}^{\mathrm{tr}}$. In order for this predictor $V_{f}$ to generalize well to slots in the same frame *f* outside the training set, it is necessary to have a sufficiently large number of training slots, $L^{\mathrm{tr}}$ [48,49].

### 2.4. Transfer Learning and Meta-Learning

In conventional learning, the number of required training slots $L^{\mathrm{tr}}$ can be reduced by selecting hyperparameters in the learning problem that reflect prior knowledge about the prediction problem at hand. In the next sections, we will explore solutions that optimize such hyperparameters based on data received from multiple previous frames. To this end, as illustrated in Figure 2, we assume the availability of channel data collected from *F* frames received in the past. In each frame, the channel follows the model described in Section 2.2. Accordingly, data from previous frames consists of L+N+δ−1 channels $\left\{h_{1,f},\ldots ,h_{L+N+\delta -1,f}\right\}$ for some integer *L*.

By using these channels, the dataset
(9)$$\begin{array}{c} Z_{f}={\left\{\left(x_{i,f},y_{i,f}\right)\right\}}_{i=1}^{L}\equiv {\left\{\left(\mathrm{vec}\left(H_{l,f}^{N}\right),h_{l+\delta ,f}\right)\right\}}_{l=N}^{L+N-1} \end{array}$$
can be obtained as explained in Section 2.1, where *L* is typically larger than $L^{\mathrm{tr}}$, although this will not be assumed in the analysis. Correspondingly, we also define the L×N input matrix $X_{f}$ and the L×1 target vector $y_{f}$. We will propose methods that leverage the historical knowledge available from dataset $Z_{f}$ for f=1,…,F via transfer learning and meta-learning with the goal of reducing number of pilots, $L^{\mathrm{tr}}$, needed for channel prediction in a new frame (i.e., frame F+1 in Figure 2).

### 2.5. Incorporating Estimation Noise

Until now, we assumed that channel vectors $h_{l,f}$ are available noiselessly to the predictor. In practice, channel information needs to be estimated via pilots. To elaborate on this point, let us assume the received signal model
(10)$$\begin{array}{c} \begin{array}{c} y_{l,f}^{p}\left[i\right]=h_{l,f}x_{l,f}^{p}\left[i\right]+n_{l,f}\left[i\right], \end{array} \end{array}$$ where $x_{l,f}^{p}\left[i\right]$ stands for the *i*th transmitted pilot symbol in block *l* of frame *f*, $y_{l,f}^{p}\left[i\right]$ for the corresponding received signal, and $h_{l,f}$ for the channel with additive white complex Gaussian noise $n_{l,f}\left[i\right]∼\mathrm{CN}\left(0,N_{0}I_{S}\right)$. Given an average energy constraint $E\left[x_{l,f}^{p}{\left[i\right]}^{2}\right]=E_{x}$ for the training symbol, the average signal-to-noise ratio (SNR) is given as $E_{x}/N_{0}$. From (Equation 10), we can estimate the channel as
(11)$$\begin{array}{c} {\overset{ˇ}{h}}_{l,f}=\frac{y_{l,f}^{p}\left[i\right]}{x_{l,f}^{p}\left[i\right]}=h_{l,f}+\frac{n_{l,f}\left[i\right]}{x_{l,f}^{p}\left[i\right]}=h_{l,f}+\xi , \end{array}$$
which suffers from channel estimation noise $\xi ∼\mathrm{CN}(0,{\mathrm{SNR}}^{-1}I_{S})$. If *P* training symbols are available in each block, the channel estimation noise can be reduced via averaging to ${\mathrm{SNR}}^{-1}/P$. Channels ${\overset{ˇ}{h}}_{l,f}$ can be used as training data in the schemes described in the previous subsections. More efficient channel estimation methods, including sparse Bayesian learning [50] and approximate message passing approaches [51] may further reduce the channel estimation noise.

## 3. Single-Antenna Frequency-Flat Channels

In this section, we propose transfer learning and meta-learning methods for single-antenna flat-fading channels, which result in S=1. Throughout this section, we write the prediction matrix $V_{f}\in C^{SN\times S}$ in (Equation 2) as the vector $v_{f}\in C^{N\times 1}$, and the target data $Y_{f}^{\mathrm{tr}}\in C^{L^{\mathrm{tr}}\times S}$ as the vector $y_{f}^{\mathrm{tr}}\in C^{L^{\mathrm{tr}}\times 1}$. Correspondingly, we rewrite the linear predictor (Equation 2) as
(12)$$\begin{array}{c} {\hat{h}}_{l+\delta ,f}=v_{f}^{†}\mathrm{vec}\left(H_{l,f}^{N}\right). \end{array}$$

### 3.1. Conventional Learning

Assuming the standard quadratic loss, we formulate the supervised learning problem as the ridge regression optimization
(13)$$\begin{array}{c} v^{*}\left(Z_{f}^{\mathrm{tr}}|\bar{v}\right)=\underset{v_{f}\in C^{N\times 1}}{arg min}\left\{\left|\right|X_{f}^{\mathrm{tr}}v_{f}-y_{f}^{\mathrm{tr}}{\left|\right|}^{2}+\lambda ||v_{f}-\bar{v}{\left|\right|}^{2}\right\}, \end{array}$$
with hyperparameters $\left(\lambda ,\bar{v}\right)$ given by the scalar λ>0 and by the N×1 bias vector $\bar{v}$. The bias vector $\bar{v}$ can be thought of defining the prior mean of the predictor $v_{f}$, whereas λ>0 specifies the precision (i.e., inverse of the variance) of this prior knowledge. The solution of problem (Equation 13) can be obtained explicitly as
(14)$$\begin{array}{c} v^{*}\left(Z_{f}^{\mathrm{tr}}|\bar{v}\right)={\left(A_{f}^{\mathrm{tr}}\right)}^{-1}\left({\left(X_{f}^{\mathrm{tr}}\right)}^{†}y_{f}^{\mathrm{tr}}+\lambda \bar{v}\right),\mathrm{with}A_{f}^{\mathrm{tr}}={\left(X_{f}^{\mathrm{tr}}\right)}^{†}X_{f}^{\mathrm{tr}}+\lambda I. \end{array}$$

### 3.2. Transfer Learning

Transfer learning uses datasets $Z_{f}$ in (Equation 9) from the previous *F* frames, i.e., with f=1,…,F, to optimize the hyperparameter vector $\bar{v}$ in (Equation 13) as
(15)$$\begin{array}{cc} {\bar{v}}^{\mathrm{trans}} & =\underset{v\in C^{N\times 1}}{arg min}\left\{\sum_{f=1}^{F}{∥X_{f}v-y_{f}∥}^{2}\right\}. \end{array}$$ The rationale for this choice is that vector ${\bar{v}}^{\mathrm{trans}}$ provides a useful prior mean to be used in the ridge regression problem (Equation 13), because it corresponds to an optimized predictor for the previous frames. Having optimized the bias vector ${\bar{v}}^{\mathrm{trans}}$, we train a channel predictor *v* via ridge regression (Equation 13) by using the training data $Z_{f^{\mathrm{new}}}$ for a new frame $f^{\mathrm{new}}$ with $L^{\mathrm{new}}$ training samples, to obtain
(16)$$\begin{array}{c} v_{f^{\mathrm{new}}}^{*}=v^{*}\left(Z_{f^{\mathrm{new}}}|{\bar{v}}^{\mathrm{trans}}\right). \end{array}$$ Note that during deployment time, this approach has the same computational complexity as that of conventional learning, because the bias vector is treated as a constant vector.

### 3.3. Meta-Learning

Unlike transfer learning, which utilizes all the available datasets ${\left\{Z_{f}\right\}}_{f=1}^{F}$ from the previous frames at once as in (Equation 15), meta-learning allows for the separate adaptation of the predictor in each frame. To this end, for each frame *f*, we split the *L* data points into $L^{\mathrm{tr}}$ training pairs ${\left\{\left(x_{i,f},y_{i,f}\right)\right\}}_{i=1}^{L^{\mathrm{tr}}}\equiv {\left\{\left(x_{i,f}^{\mathrm{tr}},y_{i,f}^{\mathrm{tr}}\right)\right\}}_{i=1}^{L^{\mathrm{tr}}}=Z_{f}^{\mathrm{tr}}$ and $L^{\mathrm{te}}=L-L^{\mathrm{tr}}$ test pairs ${\left\{\left(x_{i,f},y_{i,f}\right)\right\}}_{i=L^{\mathrm{tr}}+1}^{L}\equiv {\left\{\left(x_{i,f}^{\mathrm{te}},y_{i,f}^{\mathrm{te}}\right)\right\}}_{i=1}^{L^{\mathrm{te}}}=Z_{f}^{\mathrm{te}}$, resulting in two separate datasets, $Z_{f}^{\mathrm{tr}}$ and $Z_{f}^{\mathrm{te}}$. We correspondingly define the $L^{\mathrm{tr}}\times N$ input matrix $X_{f}^{\mathrm{tr}}$ and the $L^{\mathrm{tr}}\times 1$ target vector $y_{f}^{\mathrm{tr}}$, as well as the $L^{\mathrm{te}}\times N$ input matrix $X_{f}^{\mathrm{te}}$ and the $L^{\mathrm{te}}\times 1$ target vector $y_{f}^{\mathrm{te}}$.

The hyperparameter vector $\bar{v}$ is then optimized by minimizing the sum loss of the predictors $v^{*}\left(Z_{f}^{\mathrm{tr}}|\bar{v}\right)$ in (Equation 13) that are adapted separately for each frame f=1,…,F given the bias vector $\bar{v}$. Accordingly, estimating the loss in each frame *f* via the test set $Z_{f}^{\mathrm{te}}$ yields the meta-learning problem
(17)$$\begin{array}{cc} {\bar{v}}^{\mathrm{meta}} & =\underset{\bar{v}\in C^{N\times 1}}{arg min}\left\{\sum_{f=1}^{F}|v^{*}{\left(Z_{f}^{\mathrm{tr}}|\bar{v}\right)}^{†}x_{i,f}^{\mathrm{te}}-y_{i,f}^{\mathrm{te}}{|}^{2}\right\}. \end{array}$$

As studied in [52], the minimization in (Equation 17) is a least squares problem that can be solved in closed form as
(18)$$\begin{array}{cc} {\bar{v}}^{\mathrm{meta}} & =\underset{\bar{v}\in C^{N\times 1}}{arg min}\sum_{f=1}^{F}{∥{\tilde{X}}_{f}^{\mathrm{te}}\bar{v}-{\tilde{y}}_{f}^{\mathrm{te}}∥}^{2}\\ \; & ={\left({\tilde{X}}^{†}\tilde{X}\right)}^{-1}{\tilde{X}}^{†}\tilde{y}, \end{array}$$
where $L^{\mathrm{te}}\times N$ matrix ${\tilde{X}}_{f}^{\mathrm{te}}$ contains by row the Hermitian transpose of the N×1 pre-conditioned input vectors ${\left\{\lambda {\left(A_{f}^{\mathrm{tr}}\right)}^{-1}x_{i,f}^{\mathrm{te}}\right\}}_{i=1}^{L^{\mathrm{te}}}$, with $A_{f}^{\mathrm{tr}}={\left(X_{f}^{\mathrm{tr}}\right)}^{†}X_{f}^{\mathrm{tr}}+\lambda I$,; ${\tilde{y}}_{f}^{\mathrm{te}}$ is $L^{\mathrm{te}}\times 1$ vector containing vertically the complex conjugate of the transformed outputs $\{{\left(y_{i,f}^{\mathrm{te}}-{\left(y_{f}^{\mathrm{tr}}\right)}^{†}X_{f}^{\mathrm{tr}}{\left(A_{f}^{\mathrm{tr}}\right)}^{-1}x_{i,f}^{\mathrm{te}}\right\}}_{i=1}^{L^{\mathrm{te}}}$, the $FL^{\mathrm{te}}\times N$ matrix $\tilde{X}={\left[{\tilde{X}}_{1}^{\mathrm{te}},\ldots ,{\tilde{X}}_{F}^{\mathrm{te}}\right]}^{⊤}$ stacks vertically the $L^{\mathrm{te}}\times N$ matrices ${\left\{{\tilde{X}}_{f}^{\mathrm{te}}\right\}}_{f=1}^{F}$, and the $FL^{\mathrm{te}}\times 1$ vector $\tilde{y}={\left[{\tilde{y}}_{1}^{\mathrm{te}},\ldots ,{\tilde{y}}_{F}^{\mathrm{te}}\right]}^{⊤}$ stacks vertically the $L^{\mathrm{te}}\times 1$ vectors ${\left\{{\tilde{y}}_{f}^{\mathrm{te}}\right\}}_{f=1}^{F}$. Unlike standard meta-learning algorithms used by most papers on communications [25,29,30,32,33,34], the proposed meta-learning procedure adopts linear models, significantly reducing the computational complexity of meta-learning [52].

After meta-learning, similar to transfer learning, based on the meta-learned hyperparameter ${\bar{v}}_{\lambda }^{\mathrm{meta}}$, we train a channel predictor via ridge regression (Equation 13), obtaining
(19)$$\begin{array}{c} v_{f^{\mathrm{new}}}^{*}=v^{*}\left(Z_{f^{\mathrm{new}}}|{\bar{v}}^{\mathrm{meta}}\right). \end{array}$$

## 4. Multi-Antenna Frequency-Selective Channels

In this section, we study the more general scenario with any number of antennas and with frequency-selective channels, resulting in S>1. As we will discuss, a naïve extension of the techniques presented in the previous sections is undesirable, because this would not leverage the structure of the channel model (Equation 4). For this reason, in the following, we will introduce novel hybrid model- and data-driven solutions that build on the channel model (Equation 4).

### 4.1. Naïve Extension

We start by briefly presenting the direct extension of the approaches studied in the previous section to any S>1. Unlike the previous section, we adopt the general matrix notation introduced in Section 2. First, with S=1, conventional learning obtains the predictor by solving problem (Equation 13), which is generalized to any S>1 as the minimization
(20)$$\begin{array}{c} V^{*}\left(Z_{f}^{\mathrm{tr}}|\bar{V}\right)=\underset{V_{f}\in C^{SN\times S}}{arg min}\left\{\left|\right|X_{f}^{\mathrm{tr}}V_{f}-Y_{f}^{\mathrm{tr}}{\left|\right|}_{F}^{2}+\lambda ||V_{f}-\bar{V}{\left|\right|}_{F}^{2}\right\} \end{array}$$
over the linear prediction matrix $V_{f}$ in (Equation 2). Similarly, transfer learning computes the bias matrix ${\bar{V}}^{\mathrm{trans}}$ by solving the following generalization of problem (Equation 15),
(21)$$\begin{array}{cc} {\bar{V}}^{\mathrm{trans}} & =\underset{V\in C^{SN\times S}}{arg min}\left\{\sum_{f=1}^{F}{∥X_{f}V-Y_{f}∥}_{F}^{2}\right\}, \end{array}$$
followed by the evaluation of the predictor $V^{*}\left(Z_{f}^{\mathrm{tr}}|{\bar{V}}^{\mathrm{trans}}\right)$ using (Equation 20); whereas meta-learning addresses the following generalization of minimization (Equation 17),
(22)$$\begin{array}{cc} {\bar{V}}^{\mathrm{meta}} & =\underset{\bar{V}\in C^{SN\times S}}{arg min}\left\{\sum_{f=1}^{F}\sum_{i=1}^{L^{\mathrm{te}}}|V^{*}{\left(Z_{f}^{\mathrm{tr}}|\bar{V}\right)}^{†}x_{i,f}^{\mathrm{te}}-y_{i,f}^{\mathrm{te}}{|}^{2}\right\}, \end{array}$$
over the bias matrix $\bar{V}\in C^{SN\times S}$, which is used to compute the predictor $V^{*}\left(Z_{f}^{\mathrm{tr}}|{\bar{V}}^{\mathrm{meta}}\right)$ in (Equation 20).

The issue with the naïve extensions (Equation 21) and (Equation 22) is that the dimension of the predictor *V* and of the hyperparameter matrix $\bar{V}$ can become extremely large when *S* grows. This, in turn, may lead to overfitting in the hyperparameter space [53] when the number of frames, *F*, is limited. This form of overfitting may prevent transfer learning and meta-learning from effectively reducing the sample complexity for problem (Equation 20), because the optimized hyperparameter matrix $\bar{V}$ would be excessively dependent on the data received in the *F* previous frames. To solve this problem, we propose next to utilize the structure of the channel model (Equation 4) in order to reduce the dimension of the channel parametrization.

### 4.2. LSTD Channel Model

The channel model (Equation 4) implies that the channel vector $h_{l,f}$ in (Equation 7) and (Equation 8) can be written as the product of a frame-dependent $N_{R}N_{T}W\times D$ matrix $T_{f}$ and of a slot-dependent D×1 vector $\beta_{l,f}$ as in [41],  
(23)$$\begin{array}{c} h_{l,f}=T_{f}\beta_{l,f}, \end{array}$$
where $T_{f}$ collects space-time signatures of the *D* paths as
(24)$$\begin{array}{c} T_{f}=\left[\Omega_{1,f}^{1/2}g\left(\tau_{1,f}\right)⊗\mathrm{vec}\left(a_{1,f}\right),\ldots ,\Omega_{D,f}^{1/2}g\left(\tau_{D,f}\right)⊗\mathrm{vec}\left(a_{D,f}\right)\right], \end{array}$$
with $g\left(\tau_{d,f}\right)={\left[g\left(-\tau_{d,f}\right),\ldots ,g\left(\left(W-1\right)T-\tau_{d,f}\right)\right]}^{⊤}$ being the W×1 vector that collects the Nyquist-rate samples of the delayed waveform $g(\tau -\tau_{d,f})$, and the D×1 fading amplitude vector being defined as $\beta_{l,f}={\left[\exp\left(-jw_{1,f}t_{l}\right),\ldots ,\exp\left(-jw_{D,f}t_{l}\right)\right]}^{⊤}$.

The frame-dependent matrix $T_{f}$ is typically rank-deficient, because paths are generally not all resolvable [54,55]. To account for this structural property of the channel, as in [41], we introduce a $N_{R}N_{T}W\times K$ full-rank unitary matrix $B_{f}$, such that $\mathrm{span}\left\{T_{f}\right\}=\mathrm{span}\left\{B_{f}\right\}$ and redefine (Equation 23) as
(25)$$\begin{array}{c} h_{l,f}=B_{f}d_{l,f}. \end{array}$$ As an example, the unitary matrix $B_{f}$ can be obtained from the singular value decomposition of matrix $T_{f}$, i.e., $T_{f}=B_{f}\Lambda_{f}^{1/2}U_{f}^{†}$, by introducing the K×1 vector $d_{l,f}=\Lambda_{f}^{1/2}U_{f}^{†}\beta_{l,f}$ [41]. For future reference, we also rewrite (Equation 25) as
(26)$$\begin{array}{c} h_{l,f}=\sum_{k=1}^{K}b_{f}^{k}d_{l,f}^{k}, \end{array}$$
where $d_{l,f}^{k}$ is the *k*-th element of the vector $d_{l,f}$ and $b_{f}^{k}$ is the *k*-th column of the matrix $B_{f}$.

We will refer to matrix $B_{f}$ in (Equation 26) as the long-term space-time feature matrix, or feature matrix for short, whereas vector $d_{l,f}$ will be referred as the short-term corresponding amplitude vector. Parametrization (Equation 25) and (Equation 26) are particularly efficient when the feature matrix $B_{f}$ can be accurately estimated from the available data. For conventional learning, this requires observing a sufficiently large number of slots per frame, i.e., a large $L^{\mathrm{new}}$ [41], as well as a channel that varies sufficiently quickly across each frame. In contrast, as we will explore, transfer and meta-learning can potentially leverage data from multiple frames in order to enhance the estimation of the feature matrix.

### 4.3. LSTD-Based Prediction Model

Given the LSTD channel model (Equation 25) and (Equation 26), in this subsection we redefine the problem of predicting channel $h_{l+\delta ,f}=B_{f}d_{l+\delta ,f}$ as the problem of estimating the feature matrix $B_{f}$ and predicting the amplitude vector $d_{l+\delta ,f}$ based on the available data. This will lead to a reduced-rank parametrization of the linear predictor (Equation 2).

To start, we write the predicted channel ${\hat{h}}_{l+\delta ,f}$ as
(27)$$\begin{array}{c} {\hat{h}}_{l+\delta ,f}={\hat{B}}_{f}{\hat{d}}_{l+\delta ,f}, \end{array}$$
where ${\hat{B}}_{f}$ and ${\hat{d}}_{l+\delta ,f}$ are the estimated feature matrix and the predicted amplitude vector, respectively. To define the corresponding predictor, we first observe that the input matrix $H_{l,f}^{N}$ in (Equation 1) can be expressed by using (Equation 25) as
(28)$$\begin{array}{c} H_{l,f}^{N}=B_{f}\left[d_{l,f},\ldots ,d_{l-N+1,f}\right]. \end{array}$$ Assume now that we have an estimated feature matrix ${\hat{B}}_{f}$. If this estimate is sufficiently accurate, the *N* past amplitudes $\left[d_{l,f},\ldots ,d_{l-N+1,f}\right]\in C^{K\times N}$ can be in turn estimated from $H_{l,f}^{N}$ as
(29)$$\begin{array}{c} \left[{\hat{d}}_{l,f},\ldots ,{\hat{d}}_{l-N+1,f}\right]={\hat{B}}_{f}^{†}H_{l,f}^{N}. \end{array}$$ Consider now the prediction of the *k*-th amplitude $d_{l+\delta ,f}^{k}$. Generalizing (Equation 12), we adopt the linear predictor
(30)$$\begin{array}{c} {\hat{d}}_{l+\delta ,f}^{k}={\left(v_{f}^{k}\right)}^{†}\mathrm{vec}\left(\left[{\hat{d}}_{l,f}^{k},\ldots ,{\hat{d}}_{l-N+1,f}^{k}\right]\right), \end{array}$$
where $v_{f}^{k}$ is an N×1 prediction vector, and
(31)$$\begin{array}{c} \left[{\hat{d}}_{l,f}^{k},\ldots ,{\hat{d}}_{l-N+1,f}^{k}\right]={\left({\hat{b}}_{f}^{k}\right)}^{†}H_{l,f}^{N}\in C^{1\times N} \end{array}$$
is the *k*-th row of the matrix (Equation 29), which represents the past *N* fading scalar amplitudes that correspond to the *k*-th feature $b_{f}^{k}$. Plugging the prediction (Equation 30) into (Equation 27) yields the predicted channel ${\hat{h}}_{l+\delta ,f}$ (cf. (Equation 26))
(32)$$\begin{array}{c} {\hat{h}}_{l+\delta ,f}=\sum_{k=1}^{K}{\hat{b}}_{f}^{k}{\hat{d}}_{l+\delta ,f}^{k}. \end{array}$$

As detailed in Appendix A, inserting (Equation 30) and (Equation 31) to (Equation 32), we can express the LSTD-based prediction (Equation 32) in the form (Equation 2) as
(33)$$\begin{array}{c} {\hat{h}}_{l+\delta ,f}={\left(V_{f}^{\left(K\right)}\right)}^{†}\mathrm{vec}\left(H_{l,f}^{N}\right), \end{array}$$
where the LSTD-based predictor matrix $V_{f}^{\left(K\right)}\in C^{SN\times S}$ is given as
(34)$$\begin{array}{c} V_{f}^{\left(K\right)}=\sum_{k=1}^{K}v_{f}^{k}⊗\left({\hat{b}}_{f}^{k}{\left({\hat{b}}_{f}^{k}\right)}^{†}\right), \end{array}$$
where ⊗ is the Kronecker product. Overall illustration of LSTD-based channel prediction is summarized in Figure 3.

The LSTD prediction model (Equation 33) reduces the dimension of the learnable parameters from $S^{2}N$ (for $V_{f}$) to (S+N)K (for $V_{f}^{\left(K\right)}$). This complexity reduction comes at a minimal cost in terms of bias as long as the number of total features *K* is accurately chosen (a detailed discussion can be found in Section 4.7) and correlations across amplitudes $d_{l,f}^{k}$ for different features k=1,…,K are negligible.

### 4.4. Conventional Learning for LSTD-Based Prediction

In conventional learning, the goal is to optimize the LSTD-based predictor $V_{f}^{\left(K\right)}$ by optimizing the feature matrix ${\hat{B}}_{f}$ and the feature-wise predictors ${\left\{v_{f}^{k}\right\}}_{k=1}^{K}$ based on the available training dataset $Z_{f}^{\mathrm{tr}}$. Substituting $V_{f}$ with $V_{f}^{\left(K\right)}$ defined in (Equation 34) into the naïve extension of conventional learning in (Equation 20) yields the problem
(35)$$\begin{array}{cc} V^{(K),*}\left(Z_{f}^{\mathrm{tr}}|{\bar{V}}^{\left(K\right)}\right)=\; & \underset{\begin{array}{c} {\hat{B}}_{f},v_{f}^{1},\ldots ,v_{f}^{K}\\ V_{f}^{\left(K\right)}=\sum_{k=1}^{K}v_{f}^{k}⊗\left({\hat{b}}_{f}^{k}{\left({\hat{b}}_{f}^{k}\right)}^{†}\right) \end{array}}{arg min}\;||X_{f}^{\mathrm{tr}}V_{f}^{\left(K\right)}-Y_{f}^{\mathrm{tr}}{\left|\right|}_{F}^{2}+\lambda {∥V_{f}-{\bar{V}}^{\left(K\right)}∥}_{F}^{2},\\ \; & \;\;\;\;\;\;\;\;\;\;\;\;\;\;\mathrm{subject}\;\mathrm{to}\;{\hat{B}}_{f}^{†}{\hat{B}}_{f}=I_{K}, \end{array}$$
over the optimization variables $\left({\hat{B}}_{f},{\left\{v_{f}^{k}\right\}}_{k=1}^{K}\right)$. In (Equation 35), the hyperparameters $\left(\lambda ,{\bar{V}}^{\left(K\right)}\right)$ are given by the scalar λ>0 and by the SN×S LSTD-based bias matrix ${\bar{V}}^{\left(K\right)}$ defined as (cf. (Equation 34))
(36)$$\begin{array}{c} {\bar{V}}^{\left(K\right)}=\sum_{k=1}^{K}{\bar{v}}^{k}⊗\left({\bar{b}}^{k}{\left({\bar{b}}^{k}\right)}^{†}\right). \end{array}$$ Because the Euclidean norm regularization ${∥V_{f}-{\bar{V}}^{\left(K\right)}∥}_{F}^{2}$ in (Equation 35) mixes long-term and short-term dependencies due to (Equation 34) and (Equation 36), we propose the modification of problem (Equation 35)
(37)$$\begin{array}{cc} V^{(K),*}\left(Z_{f}^{\mathrm{tr}}|{\left\{{\bar{b}}^{k},{\bar{v}}^{k}\right\}}_{k=1}^{K}\right) & =\;\underset{\begin{array}{c} {\hat{B}}_{f},v_{f}^{1},\ldots ,v_{f}^{K}\\ V_{f}^{\left(K\right)}=\sum_{k=1}^{K}v_{f}^{k}⊗\left({\hat{b}}_{f}^{k}{\left({\hat{b}}_{f}^{k}\right)}^{†}\right) \end{array}}{arg min}\;\{||X_{f}^{\mathrm{tr}}V_{f}^{\left(K\right)}-Y_{f}^{\mathrm{tr}}{\left|\right|}_{F}^{2}+\lambda_{2}\sum_{k=1}^{K}\left|\right|v_{f}^{k}-{\bar{v}}^{k}{\left|\right|}^{2}\\ & \;\;\;\;\;\;\;\;\;\;\;-\lambda_{1}\sum_{k=1}^{K}\mathrm{tr}\left({\left({\hat{b}}_{f}^{k}\right)}^{†}\left({\bar{b}}^{k}{\left({\bar{b}}^{k}\right)}^{†}\right){\hat{b}}_{f}^{k}\right)\},\\ \; & \;\;\;\;\;\;\;\;\;\;\;\;\;\;\mathrm{subject}\;\mathrm{to}\;{\hat{B}}_{f}^{†}{\hat{B}}_{f}=I_{K}, \end{array}$$ with hyperparameters $\left(\lambda_{1},\lambda_{2},{\bar{b}}^{1},\ldots ,{\bar{b}}^{K},{\bar{v}}^{1},\ldots ,{\bar{v}}^{K}\right)$ given by the scalars $\lambda_{1},\lambda_{2}>0$, by the S×1 long-term bias vectors ${\bar{b}}^{1},\ldots ,{\bar{b}}^{K}$, and by the N×1 short-term bias vectors ${\bar{v}}^{1},\ldots ,{\bar{v}}^{K}$. For each feature *k*, the considered regularization minimizes the Euclidean distance between the short-term prediction vector $v_{f}^{k}$ and the short-term bias vector ${\bar{v}}^{k}$ as in Section 3, while maximizing the alignment between the long-term feature vector ${\hat{b}}_{f}^{k}$ and the long-term bias vector ${\bar{b}}^{k}$ in a manner akin to the kernel alignment method of [56].

To address problem (Equation 37), inspired by [57,58], we propose a sequential approach, in which the pair $\left(v_{f}^{k},{\hat{b}}_{f}^{k}\right)$ consisting of the *k*-th predictor $v_{f}^{k}$ and the *k*-th feature vector ${\hat{b}}_{f}^{k}$ is optimized in the order k=1,2,…,K. Specifically, at each step *k*, we consider the problem
(38)$$\begin{array}{cc} {\hat{b}}_{f}^{k,*},v_{f}^{k,*}=\;\underset{\begin{array}{c} {\hat{b}}_{f}^{k},v_{f}^{k}\\ {\left(V_{f}^{\left(K\right)}\right)}^{k}=v_{f}^{k}⊗\left({\hat{b}}_{f}^{k}{\left({\hat{b}}_{f}^{k}\right)}^{†}\right) \end{array}}{arg min}\;\{||X_{f}^{\mathrm{tr}}{\left(V_{f}^{\left(K\right)}\right)}^{k}-{\left(Y_{f}^{\mathrm{tr}}\right)}^{k}{\left|\right|}_{F}^{2} & -\lambda_{1}\mathrm{tr}\left({\left({\hat{b}}_{f}^{k}\right)}^{†}\left({\bar{b}}^{k}{\left({\bar{b}}^{k}\right)}^{†}\right){\hat{b}}_{f}^{k}\right)+\lambda_{2}\left|\right|v_{f}^{k}-{\bar{v}}^{k}{\left|\right|}^{2}\},\\ \; & \;\;\;\;\;\;\mathrm{subject}\;\mathrm{to}\;{\left({\hat{b}}_{f}^{k}\right)}^{†}{\hat{b}}_{f}^{k}=1, \end{array}$$
where the $L^{\mathrm{tr}}\times S$*k*-th residual target matrix ${\left(Y_{f}^{\mathrm{tr}}\right)}^{k}$ is defined as [57,58]
(39)$$\begin{array}{l} {\left(Y_{f}^{\mathrm{tr}}\right)}^{k}=\left\{\begin{array}{l} Y_{f}^{\mathrm{tr}},\;\mathrm{for}\;k=1,\\ Y_{f}^{\mathrm{tr}}-\sum_{k^{\prime }=1}^{k-1}X_{f}^{\mathrm{tr}}{\left(V_{f}^{\left(K\right)}\right)}^{k^{\prime },*},\;\mathrm{for}\;k>1, \end{array}\right. \end{array}$$
given the *k*-th predictor
(40)$$\begin{array}{c} {\left(V_{f}^{\left(K\right)}\right)}^{k}=v_{f}^{k}⊗\left({\hat{b}}_{f}^{k}{\left({\hat{b}}_{f}^{k}\right)}^{†}\right) \end{array}$$
and *k*-th optimized predictor
(41)$$\begin{array}{c} {\left(V_{f}^{\left(K\right)}\right)}^{k,*}=v_{f}^{k,*}⊗\left({\hat{b}}_{f}^{k,*}{\left({\hat{b}}_{f}^{k,*}\right)}^{†}\right). \end{array}$$

Because (Equation 38) is a nonconvex problem, we consider alternating least squares (ALS) [59] to obtain the optimal solution $\left\{{\hat{b}}_{f}^{k,*},v_{f}^{k,*}\right\}$ by iterating between the following steps: (i) for a fixed ${\hat{b}}_{f}^{k}$, update $v_{f}^{k}$ as
(42)$$\begin{array}{c} v_{f}^{k}\leftarrow \;\underset{\begin{array}{c} v_{f}^{k}\\ {\left(V_{f}^{\left(K\right)}\right)}^{k}=v_{f}^{k}⊗{\left({\hat{b}}_{f}^{k}\left({\hat{b}}_{f}^{k}\right)\right)}^{†} \end{array}}{arg min}\;\left\{\left|\right|X_{f}^{\mathrm{tr}}{\left(V_{f}^{\left(K\right)}\right)}^{k}-{\left(Y_{f}^{\mathrm{tr}}\right)}^{k}{\left|\right|}_{F}^{2}+\lambda_{2}\left|\right|v_{f}^{k}-{\bar{v}}^{k}{\left|\right|}^{2}\right\}; \end{array}$$
and (ii) for a fixed $v_{f}^{k}$, update ${\hat{b}}_{f}^{k}$ as
(43)$$\begin{array}{cc} {\hat{b}}_{f}^{k}\leftarrow \;\underset{\begin{array}{c} {\hat{b}}_{f}^{k}\\ {\left(V_{f}^{\left(K\right)}\right)}^{k}=v_{f}^{k}⊗\left({\hat{b}}_{f}^{k}{\left({\hat{b}}_{f}^{k}\right)}^{†}\right) \end{array}}{arg min}\;\{||X_{f}^{\mathrm{tr}}{\left(V_{f}^{\left(K\right)}\right)}^{k}-{\left(Y_{f}^{\mathrm{tr}}\right)}^{k}{\left|\right|}_{F}^{2} & -\lambda_{1}\mathrm{tr}\left({\left({\hat{b}}_{f}^{k}\right)}^{†}\left({\bar{b}}^{k}{\left({\bar{b}}^{k}\right)}^{†}\right){\hat{b}}_{f}^{k}\right)\},\\ \; & \;\;\;\mathrm{subject}\;\mathrm{to}\;{\left({\hat{b}}_{f}^{k}\right)}^{†}{\hat{b}}_{f}^{k}=1, \end{array}$$
until convergence. Closed-form solutions for (Equation 42) and (Equation 43) can be found in Appendix B, and the overall LSTD-based conventional learning scheme can be found in Algorithm 1.
**Algorithm 1:** LSTD-based conventional learning for channel prediction for S≥1
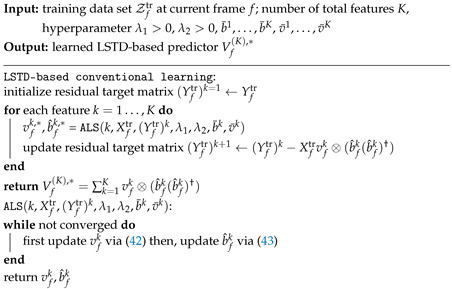


### 4.5. Transfer Learning for LSTD-Based Prediction

Similar to conventional learning, transfer learning for LSTD-based prediction can be addressed from the naïve extension (Equation 21) by utilizing the LSTD parametrization $V^{\left(K\right)}$ in (Equation 34) in lieu of the unconstrained predictor *V* to obtain the bias matrix ${\bar{V}}^{(K),\mathrm{trans}}$ as
(44)$$\begin{array}{c} {\bar{V}}^{(K),\mathrm{trans}}=\;\underset{\begin{array}{c} \hat{B},v^{1},\ldots ,v^{K}\\ V^{\left(K\right)}=\sum_{k=1}^{K}v^{k}⊗\left({\hat{b}}^{k}{\left({\hat{b}}^{k}\right)}^{†}\right) \end{array}}{arg min}\;\left\{\sum_{f=1}^{F}\left|\right|X_{f}V^{\left(K\right)}-Y_{f}{\left|\right|}_{F}^{2}\right\},\\ \mathrm{subject}\;\mathrm{to}\;{\hat{B}}^{†}\hat{B}=I_{K}, \end{array}$$
which can also be solved via the ALS-based sequential approach detailed in Section 4.4. This produces the sequences ${\bar{b}}^{1,\mathrm{trans}},\ldots ,{\bar{b}}^{K,\mathrm{trans}}$ and ${\bar{v}}^{1,\mathrm{trans}},\ldots ,{\bar{v}}^{K,\mathrm{trans}}$ as (cf. (Equation 38))
(45)$$\begin{array}{cc} {\bar{b}}^{k,\mathrm{trans}},{\bar{v}}^{k,\mathrm{trans}}=\;\underset{\begin{array}{c} {\hat{b}}^{k},v^{k}\\ {\left(V^{\left(K\right)}\right)}^{k}=v^{k}⊗\left({\hat{b}}^{k}{\left({\hat{b}}^{k}\right)}^{†}\right) \end{array}}{arg min}\; & \left\{\sum_{f=1}^{F}\left|\right|X_{f}{\left(V^{\left(K\right)}\right)}^{k}-{\left(Y_{f}\right)}^{k}{\left|\right|}_{F}^{2}\right\},\\ \; & \;\;\;\;\mathrm{subject}\;\mathrm{to}\;{\left({\hat{b}}^{k}\right)}^{†}{\hat{b}}^{k}=1, \end{array}$$
where the residual target matrix ${\left(Y_{f}\right)}^{k}$ is defined as (cf. (Equation 39))
(46)$$\begin{array}{l} {\left(Y_{f}\right)}^{k}=\left\{\begin{array}{l} Y_{f},\;\mathrm{for}\;k=1,\\ Y_{f}-\sum_{k^{\prime }=1}^{k-1}X_{f}{\left(V_{f}^{\left(K\right)}\right)}^{k^{\prime },\mathrm{trans}},\;\mathrm{for}\;k>1 \end{array}\right. \end{array}$$
with *k*-th optimized predictor
(47)$$\begin{array}{c} {\left(V^{\left(K\right)}\right)}^{k,\mathrm{trans}}={\bar{v}}^{k,\mathrm{trans}}⊗{\left({\bar{b}}^{k,\mathrm{trans}}\left({\bar{b}}^{k,\mathrm{trans}}\right)\right)}^{†}. \end{array}$$ Details for transfer learning can be found in Appendix C, and the overall transfer learning scheme for LSTD prediction is summarized in Algorithm 2. After transfer learning, similar to Section 3.2, based on the optimized hyperparameters ${\bar{b}}^{1,\mathrm{trans}},\ldots ,{\bar{b}}^{K,\mathrm{trans}}$ and ${\bar{v}}^{1,\mathrm{trans}},\ldots ,{\bar{v}}^{K,\mathrm{trans}}$, the LSTD-based channel predictor for a new frame $f^{\mathrm{new}}$ can be obtained via (Equation 37) as
(48)$$\begin{array}{c} V_{f^{\mathrm{new}}}^{(K),*}=V^{(K),*}\left(Z_{f^{\mathrm{new}}}^{\mathrm{tr}}|{\left\{{\bar{b}}^{k,\mathrm{trans}},{\bar{v}}^{k,\mathrm{trans}}\right\}}_{k=1}^{K}\right), \end{array}$$
which can also be solved in the sequential way as in (Equation 38).
**Algorithm 2:** LSTD-based transfer-learning for channel prediction for S≥1
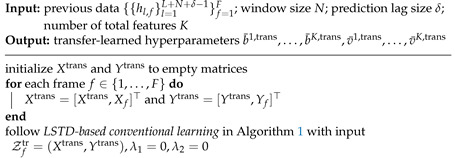


### 4.6. Meta-Learning for LSTD-Based Prediction

Plugging (Equation 37) into the naïve extension of (Equation 22), we can formulate the meta-learning problem for LSTD-based prediction as
(49)$$\begin{array}{c} \min_{{\left\{{\bar{b}}^{k},{\bar{v}}^{k}\right\}}_{k=1}^{K}}\sum_{f=1}^{F}{∥X_{f}^{\mathrm{te}}V^{(K),*}\left(Z_{f}^{\mathrm{tr}}|{\left\{{\bar{b}}^{k},{\bar{v}}^{k}\right\}}_{k=1}^{K}\right)-Y_{f}^{\mathrm{te}}∥}_{F}^{2}. \end{array}$$ Similar to the sequential approach (Equation 38) described in Section 4.4, we propose a hierarchical sequential approach for meta-learning by using (Equation 38) in the order k=1,…,K, obtaining the problem  
(50)$$\begin{array}{c} {\bar{b}}^{k,\mathrm{meta}},{\bar{v}}^{k,\mathrm{meta}}=\;\underset{\begin{array}{c} {\bar{b}}^{k},{\bar{v}}^{k}\\ {\left(V_{f}^{\left(K\right)}\right)}^{k,*}=v_{f}^{k,*}⊗\left({\hat{b}}_{f}^{k,*}{\left({\hat{b}}_{f}^{k,*}\right)}^{†}\right) \end{array}}{arg min}\;\left\{\sum_{f=1}^{F}{∥X_{f}^{\mathrm{te}}{\left(V_{f}^{\left(K\right)}\right)}^{k,*}-{\left(Y_{f}^{\mathrm{te}}\right)}^{k}∥}_{F}^{2}\right\}, \end{array}$$
with the residual target matrix ${\left(Y_{f}^{\mathrm{te}}\right)}^{k}$ defined as (cf. (Equation 39))
(51)$$\begin{array}{l} {\left(Y_{f}^{\mathrm{te}}\right)}^{k}=\left\{\begin{array}{l} Y_{f}^{\mathrm{te}},\;\mathrm{for}\;k=1,\\ Y_{f}^{\mathrm{te}}-\sum_{k^{\prime }=1}^{k-1}X_{f}^{\mathrm{te}}{\left(V_{f}^{\left(K\right)}\right)}^{k^{\prime },*},\;\mathrm{for}\;k>1. \end{array}\right. \end{array}$$

The bilevel non-convex optimization problem (Equation 50) is addressed through gradient-based updates with gradients computed via equilibrium propagation (EP) [60,61]. EP uses finite differentiation to approximate the gradient of the bilevel optimization (Equation 50), where the difference is computed between two gradients obtained at two stationary points $\left({\hat{b}}_{f}^{k,*},v_{f}^{k,*}\right)$ and $\left({\hat{b}}_{f}^{k,\alpha },v_{f}^{k,\alpha }\right)$ for the original problem (Equation 38) and modified version of (Equation 38) that considers additional prediction loss for the test set $Z_{f}^{\mathrm{te}}$. Specifically, EP leverages the asymptotic equality [60]
(52)$$\begin{array}{cc} \nabla_{{\bar{b}}^{k}} & \sum_{f=1}^{F}{∥X_{f}^{\mathrm{te}}{\left(V_{f}^{\left(K\right)}\right)}^{k,*}-{\left(Y_{f}^{\mathrm{te}}\right)}^{k}∥}_{F}^{2}=\lim_{\alpha \to 0}\frac{2\lambda_{1}}{\alpha }\sum_{f=1}^{F}\left({\hat{b}}_{f}^{k,*}{\left({\hat{b}}_{f}^{k,*}\right)}^{†}-{\hat{b}}_{f}^{k,\alpha }{\left({\hat{b}}_{f}^{k,\alpha }\right)}^{†}\right){\bar{b}}^{k} \end{array}$$
and
(53)$$\begin{array}{cc} \nabla_{{\bar{v}}^{k}}\sum_{f=1}^{F} & {∥X_{f}^{\mathrm{te}}{\left(V_{f}^{\left(K\right)}\right)}^{k,*}-{\left(Y_{f}^{\mathrm{te}}\right)}^{k}∥}_{F}^{2}=\lim_{\alpha \to 0}\frac{2\lambda_{2}}{\alpha }\sum_{f=1}^{F}\left(v_{f}^{k,*}-v_{f}^{k,\alpha }\right), \end{array}$$
with additional real-valued hyperparameter α∈R, which is generally chosen to be a non-zero small value [60,61]. In (Equation 52) and (Equation 53), vectors ${\hat{b}}_{f}^{k,\alpha }$ and $v_{f}^{k,\alpha }$ are defined as (cf. (Equation 38))
(54)$$\begin{array}{cc} {\hat{b}}_{f}^{k,\alpha },v_{f}^{k,\alpha }=\;\underset{\begin{array}{c} {\hat{b}}_{f}^{k},v_{f}^{k}\\ {\left(V_{f}^{\left(K\right)}\right)}^{k}=v_{f}^{k}⊗\left({\hat{b}}_{f}^{k}{\left({\hat{b}}_{f}^{k}\right)}^{†}\right) \end{array}}{arg min}\;\{\left|\right|X_{f}^{\mathrm{tr}}{\left(V_{f}^{\left(K\right)}\right)}^{k} & -{\left(Y_{f}^{\mathrm{tr}}\right)}^{k}{\left|\right|}_{F}^{2}+\alpha ||X_{f}^{\mathrm{te}}{\left(V_{f}^{\left(K\right)}\right)}^{k}-{\left(Y_{f}^{\mathrm{te}}\right)}^{k}{\left|\right|}_{F}^{2}\\ & -\lambda_{1}\mathrm{tr}\left({\left({\hat{b}}_{f}^{k}\right)}^{†}\left({\bar{b}}^{k}{\left({\bar{b}}^{k}\right)}^{†}\right){\hat{b}}_{f}^{k}\right)+\lambda_{2}\left|\right|v_{f}^{k}-{\bar{v}}^{k}{\left|\right|}^{2}\},\\ \; & \;\;\;\;\;\;\;\;\;\;\mathrm{subject}\;\mathrm{to}\;{\left({\hat{b}}_{f}^{k}\right)}^{†}{\hat{b}}_{f}^{k}=1. \end{array}$$ Derivations for the gradients (Equation 52) and (Equation 53) can be found in Appendix D.

To reduce the computational complexity for the gradient-based updates, we adopt stochastic gradient descent with the Adam optimizer as done in [61] in order to update ${\bar{b}}^{k}$ and ${\bar{v}}^{k}$ based on (Equation 52) and (Equation 53). The overall LSTD-based meta-learning scheme is detailed in Algorithm 3.

After meta-learning, as in Section 3.3, based on the optimized ${\bar{b}}^{1,\mathrm{meta}},\ldots ,{\bar{b}}^{K,\mathrm{meta}}$ and ${\bar{v}}^{1,\mathrm{meta}},\ldots ,{\bar{v}}^{K,\mathrm{meta}}$, LSTD-based channel predictor for a new frame $f^{\mathrm{new}}$ can be obtained via (Equation 37) as
(55)$$\begin{array}{c} V_{f^{\mathrm{new}}}^{(K),*}=V^{(K),*}\left(Z_{f^{\mathrm{new}}}^{\mathrm{tr}}|{\left\{{\bar{b}}^{k,\mathrm{meta}},{\bar{v}}^{k,\mathrm{meta}}\right\}}_{k=1}^{K}\right), \end{array}$$
which can be solved in a sequential way, as in (Equation 38).

The computational complexity order of the considered schemes is summarized in Table 1 and Table 2. At deployment time, as seen in Table 1, all schemes require the same computational complexity of conventional learning. In contrast, in the offline meta-learning or transfer learning phase, the computational overhead depends on the dimension of the channel vector $S=N_{R}N_{T}W$. LSTD-based schemes can reduce the computational overhead as compared to naïve solutions when the channel vector is large, i.e., S≫1, and the rank *K* is sufficiently small. This is quantified in Table 2, where $I_{\mathrm{ALS}}$ is the number of iterations for ALS and $I_{\mathrm{EP}}$ is the number of iterations for EP.
**Algorithm 3:** LSTD-based meta-learning for channel prediction for S≥1
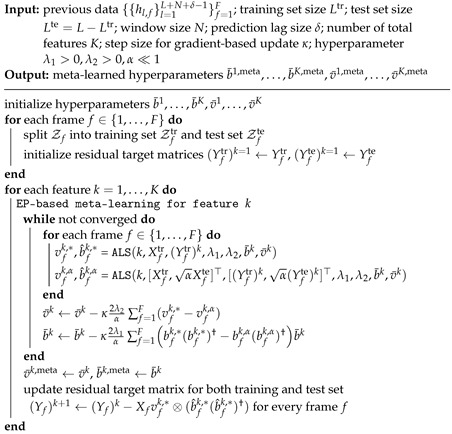


### 4.7. Rank-Estimation for LSTD-Based Prediction

The number of total features *K* for LSTD-based predictions depends on the rank of the unknown space-time signature matrix $T_{f}$ as discussed in Section 4.2. This rank can be estimated by using available channels from previous frames if we assume that the number of total features does not change over multiple frames. This can be achieved via one of the standard methods, Akaike’s information theoretic criterion (AIC) (Equation (Equation 16) in [62]), which is applicable for all the proposed LSTD-based techniques. However, as the AIC-based rank estimation generally tends to be overestimated [62,63], we propose a potentially more effective estimator for meta-learning, which utilizes a validation dataset.

To this end, we first split the available *F* frames into $F^{\mathrm{tr}}$ meta-training frames $f=1,\ldots ,F^{\mathrm{tr}}$ and $F^{\mathrm{val}}$ meta-validation frames $f=F^{\mathrm{tr}}+1,\ldots ,F$. Then, we compute the sum-loss as (cf. (Equation 49))
(56)$$\begin{array}{c} \sum_{f=F^{\mathrm{tr}}+1}^{F^{\mathrm{tr}}+F^{\mathrm{val}}}∥X_{f}^{\mathrm{te}}V^{(k),*}\left(Z_{f}^{\mathrm{tr}}|{\left\{{\bar{b}}^{k^{\prime },\mathrm{meta}},{\bar{v}}^{k^{\prime },\mathrm{meta}}\right\}}_{k^{\prime }=1}^{k}\right)-Y_{f}^{\mathrm{te}}{∥}_{F}^{2}, \end{array}$$
where the hyperparameters ${\left\{{\bar{b}}^{k^{\prime },\mathrm{meta}},{\bar{v}}^{k^{\prime },\mathrm{meta}}\right\}}_{k^{\prime }=1}^{k}$ are computed by using the $F^{\mathrm{tr}}$ meta-training frames, as explained in the previous section. The rank-estimation procedure sequentially evaluates the meta-validation loss (Equation 56) in order to minimize it over the selection of *k*. In this regard, it is worth noting that an increase in the total number of features *k* always decreases the meta-training loss in (Equation 49), whereas this is not necessarily true for the meta-validation and meta-test losses.

## 5. Experiments

In this section, we present experimental results for the prediction of multi-antenna and/or frequency-selective channels. Numerical examples for single-antenna frequency-flat channels for both offline and online learning scenarios can be found in the conference version of this paper [45]. For all the experiments, we compute the normalized mean squared error (NMSE) $\left|\right|{\hat{h}}_{l+\delta ,f}-h_{l+\delta ,f}{\left|\right|}^{2}/\left|\right|h_{l+\delta ,f}{\left|\right|}^{2}$, which is averaged over 100 samples for 200 new frames. To avoid discrepancies between the evaluation measures used during the training and testing phase, we also adopt the NMSE as the training loss function by normalizing the training dataset for the new frame $f^{\mathrm{new}}$ as (cf. (Equation 3))
(57)$$\begin{array}{c} Z_{f^{\mathrm{new}}}^{\mathrm{tr}}={\left\{\left(x_{i,f^{\mathrm{new}}},y_{i,f^{\mathrm{new}}}\right)\right\}}_{i=1}^{L^{\mathrm{tr}}}\\ \;\;\equiv {\left\{\left(\mathrm{vec}(H_{l,f^{\mathrm{new}}}^{N})/\left|\right|h_{l+\delta ,f^{\mathrm{new}}}\left|\right|,h_{l+\delta ,f^{\mathrm{new}}}/\left|\right|h_{l+\delta ,f^{\mathrm{new}}}\left|\right|\right)\right\}}_{l=N}^{L^{\mathrm{tr}}+N-1}, \end{array}$$
and similarly redefine the datasets from previous frames f=1,…,F for transfer and meta-training as (cf. (Equation 9))
(58)$$\begin{array}{c} Z_{f}={\left\{\left(x_{i,f},y_{i,f}\right)\right\}}_{i=1}^{L}\\ \;\equiv {\left\{\left(\mathrm{vec}(H_{l,f}^{N})/\left|\right|h_{l+\delta ,f}\left|\right|,h_{l+\delta ,f}/\left|\right|h_{l+\delta ,f}\left|\right|\right)\right\}}_{l=N}^{L+N-1}. \end{array}$$

As summarized in Table 3, we consider a window size N=5 with lag size δ=3. All of the experimental results follow the 3GPP 5G standard SCM channel model [46] with variations of the long-term features over frames following Clause 7.6.3.2 (Procedure B) [46], under the Umi–Street Canyon environment, as discussed in Section 2.2. The normalized Doppler frequency $\rho =\gamma_{d,f}/\gamma_{\mathrm{SRS}}\in \left[0,1\right]$ within each frame *f*, defined as the ratio between the Doppler frequency $\gamma_{d,f}$ (Equation 4) and the frequency of the pilot symbols $\gamma_{\mathrm{SRS}}$, or sounding reference signal (SRS) [46], is randomly selected in one of the two following ways: (i) for slow-varying environments, it is uniformly drawn in the interval [0.005,0.05]; and (ii) for fast-varying environments, it is uniformly distributed in the interval [0.1,1]. In the following, we study the impact of (i) the number of antennas $N_{R}N_{T}$, (ii) the number of channel taps *W*, (iii) the number of training samples $L^{\mathrm{new}}$, and (iv) the number of previous frames *F*, for various prediction schemes: (a) conventional learning, (b) transfer learning, and (c) meta-learning, where each scheme is implemented by using either the naïve or the LSTD parametrization. We set λ=0, $\lambda_{1}=0$, and $\lambda_{2}=0$ for conventional learning [9,45], whereas λ=1, $\lambda_{1}=1$, and $\lambda_{2}=1$ for transfer and meta-learning.

### 5.1. Multi-Antenna Frequency-Flat Channels

We begin by considering multi-antenna frequency-flat channels and evaluating the NMSE as a function of total number of antennas $N_{R}N_{T}$ under a fast-varying environment (Figure 4) or a slow-varying environment (Figure 5). We set K=1 in the LSTD model. Specific antenna configurations are described in Appendix E. Both transfer and meta-learning are seen to provide significant advantages as compared to conventional learning, as long as one chooses the type of parametrization—naïve or LSTD—as a function of the type of variability in the channel, with meta-learning generally outperforming transfer learning. In particular, as seen in Figure 4, for fast-varying environments, meta-learning with LSTD parametrization has the best performance, significantly reducing the NMSE with respect to both conventional and transfer learning. This is because meta-learning with LSTD can account for the need to adapt to fast-varying channel conditions, while also leveraging the reduced-rank structure of the channel. In contrast, as shown in Figure 5, for slow-varying channels, naïve parametrization tends to be preferable, because, as explained in Section 4.2, long-term and short-term features of the channel become indistinguishable when channel variability is too low. It is also interesting to observe that increasing the number of antennas is generally useful for prediction, as the predictor can build on a larger vector of correlated covariates. This is, however, not the case for conventional learning in slow-varying environments, for which the features tend to be too correlated, resulting in overfitting. As a final note, although absolute NMSE values close to 1 may be insufficient for use in applications such as precoding, they can provide useful information for other applications such as proactive resource allocation [40,64].

### 5.2. Rank Estimation

In the previous experiments, we have considered channels with unitary rank, for which one can assume without loss of optimality a number of features in the LSTD parametrization equal to K=1. In order to implement predictors for multi-antenna frequency-selective channels, one instead needs to first address the problem of estimating the number of features. Here, we evaluate the performance of the approach proposed in Section 4.7 for rank estimation. To this end, we set the number of antennas as $N_{R}=8$ and $N_{T}=8$, and consider the 19-clustered channel model with delay spread ratio 2. Figure 6 shows the NMSE evaluated on the meta-training, meta-validation, and meta-test data sets as a function of total number of features *K*. The meta-training set contains 20 frames, the meta-test 200 frames, and the meta-validation set 20 frames. The meta-training loss is monotonically decreasing with *K*, because a richer parametrization enables a closer fit of the training data. In contrast, both meta-test and meta-validation loss are optimized for an intermediate value of *K*. The main point of the figure is that the meta-validation loss, while only containing 20 frames, provides useful information to choose a value of *K* that approximately minimizes the meta-test loss. In contrast, although we can see that K=3 is a proper estimate of the channel rank for the considered set-up, AIC-based rank estimation gives the highly overestimated value K=200, which deteriorates the prediction performance, as can be seen in Figure 6. Throughout the following experiments, we will follow the proposed procedure to select *K* for meta-learning, whereas for all the other schemes, we adopt AIC-based rank estimation to determine *K*.

### 5.3. Single-Antenna Frequency-Selective Channels

Before considering multi-antenna frequency-selective channels, we first consider the impact of the level of frequency selectivity on the prediction of single-antenna frequency-selective channels. To this end, starting from 45ns, we increase the delay spread by a multiplicative factor, and correspondingly also increase the number of taps by the same amount, which is referred to as delay spread ratio in Figure 6. The number of taps *W* is obtained as the smallest number of taps that contains more than 90% of the average channel power, following ITU-R report [65]. Figure 7 shows that the dependence on the delay spread of the channel is qualitatively similar to the dependence on the number of antennas in Figure 4 and Figure 5, with the top of Figure 7 representing the performance under a fast-varying environment and the bottom figure depicting the NMSE for a slow-varying environment. Accordingly, as discussed in the previous subsection, meta-learning outperforms both transfer and conventional learning, as long as the parametrization is correctly selected: naïve for slow-varying channels, and LSTD for fast-varying environments.

### 5.4. Multi-Antenna Frequency-Selective Channel Case

We now consider the prediction performance for multi-antenna frequency-selective channels as a function of the number of training samples $L^{\mathrm{new}}$ in Figure 8 and Figure 9, as well as versus the number of frames *F* in Figure 10. For meta-learning, we set $L^{\mathrm{tr}}=L^{\mathrm{new}}$ in order to avoid discrepancies between meta-training and meta-testing [29]. Figure 8 and Figure 9 shows that meta-learning and transfer learning, which utilize F=500 previous frames, can significantly outperform conventional learning in terms of number of required pilots $L^{\mathrm{new}}$. This key observation motivates the use of transfer and meta-learning in the presence of limited training data. Furthermore, confirming the analysis in Section 3.3 and Section 4.6, meta-learning can outperform all other schemes as long as one selects a naïve parametrization for slowly varying environments, and the LSTD parametrization for fast-varying environments. For sufficiently large $L^{\mathrm{new}}$, transfer learning can, however, improve over meta-learning on fast-varying environments, as seen in Figure 8. This stems from the split of training and testing set applied by meta-learning, which can lead to a performance loss as $L^{\mathrm{new}}$ increases.

Lastly, we investigate the effect of the number of previous frames *F* for transfer and meta-learning. As a general result, as demonstrated by Figure 10, an increase in the number *F* of previous frames results in better performance for both transfer and meta-learning. Furthermore, in a slow-varying environment with a small value of *F*, transfer learning can outperform meta-learning due to the limited need for adaptation, whereas meta-learning with the correctly select type of parametrization, outperforms transfer learning otherwise.

## 6. Conclusions

In this paper, we have introduced data-driven channel prediction strategies for multi-antenna frequency-selective channels that aim at reducing the number of pilots by integrating transfer and meta-learning with a novel parametrization of linear predictors. The methods leverage the underlying structure of the wireless channels, which can be expressed in terms of a long short-term decomposition (LSTD) into long-term space-time features and fading amplitudes. To enable transfer and meta-learning under an LSTD-based model, we have proposed an optimization strategy based on equilibrium propagation (EP) and alternating least squares (ALS). Numerical experiments have shown that the proposed LSTD-based transfer and meta-learning methods far outperform conventional prediction methods, especially in the few-pilots regime. For instance, under a standard 3GPP SCM channel model, assuming four transmit antennas and two receive antennas, using only one pilot meta-learning with LSTD can reduce the normalized prediction MSE by 3 dB as compared to standard learning techniques. Future work may consider the joint use of deep neural networks, in lieu of linear prediction filters, although related results for multi-antenna frequency-flat channels have not reported any significant advantage to date [14,15,16,19].

## Figures and Tables

**Figure 1 entropy-24-01363-f001:**
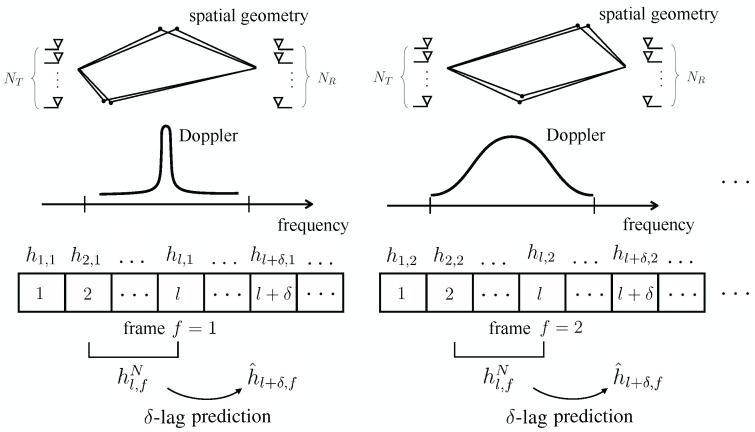
Illustration of the frame-based transmission system under study. At any frame *f*, based on the previous *N* channels $h_{l,f}^{N}$, we investigate the problem of optimizing the δ-lag prediction ${\hat{h}}_{l+\delta ,f}$.

**Figure 2 entropy-24-01363-f002:**
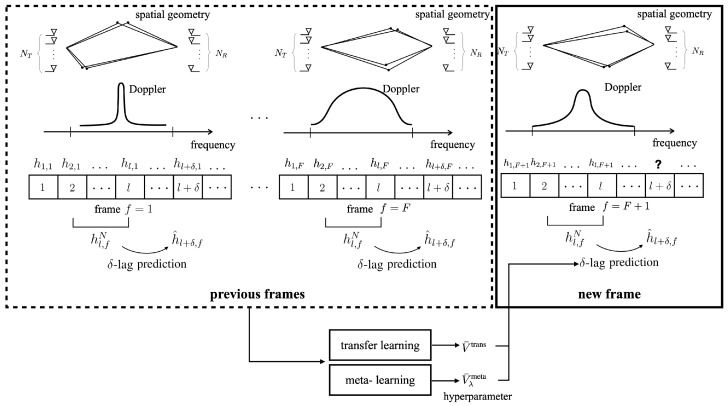
Illustration of the considered transfer and meta-learning methods. With access to pilots from previously received frames, transfer learning and meta-learning aim at obtaining the hyperparameters $\bar{V}$ to be used for channel prediction in a new frame.

**Figure 3 entropy-24-01363-f003:**
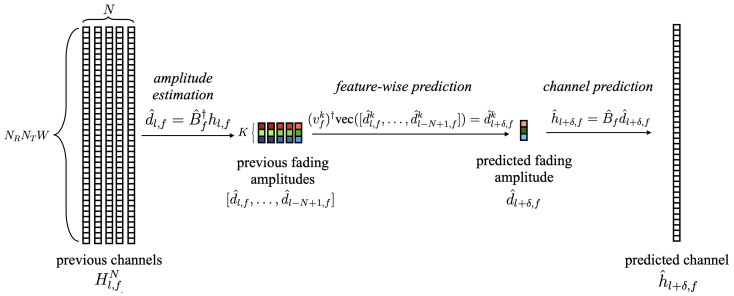
Illustration of the considered LSTD-based prediction model. (i) Estimate amplitudes $d_{l,f}$ via the estimated long-term feature matrix ${\hat{B}}_{f}$. (ii) Feature-wise short-term fading amplitude prediction ${\hat{d}}_{l+\delta ,f}^{k}$ based on feature-wise predictor $v_{f}^{k}$ for k=1,…,K. (iii) Reconstruction of the predicted channel ${\hat{h}}_{l+\delta ,f}$ based on the feature matrix ${\hat{B}}_{f}$ and predicted fading amplitude ${\hat{d}}_{l+\delta ,f}$.

**Figure 4 entropy-24-01363-f004:**
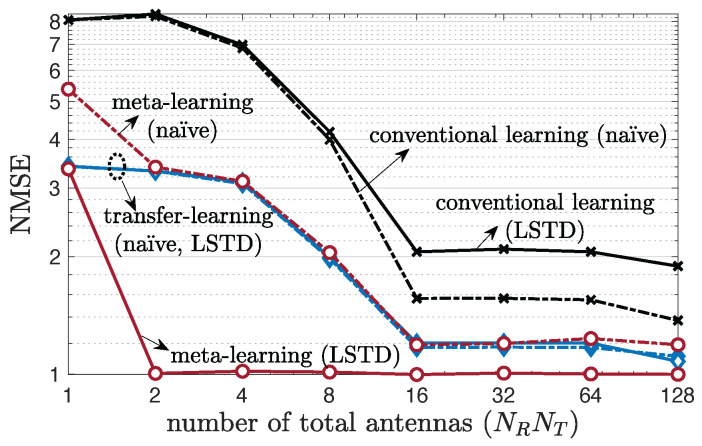
Multi-antenna frequency-flat channel prediction performance as a function of the total number of antennas, $N_{R}N_{T}$, under a single-clustered, single-tap (W=1), 3GPP SCM channel model for a fast-varying environment with number of training samples $L^{\mathrm{new}}=1$ (K=1).

**Figure 5 entropy-24-01363-f005:**
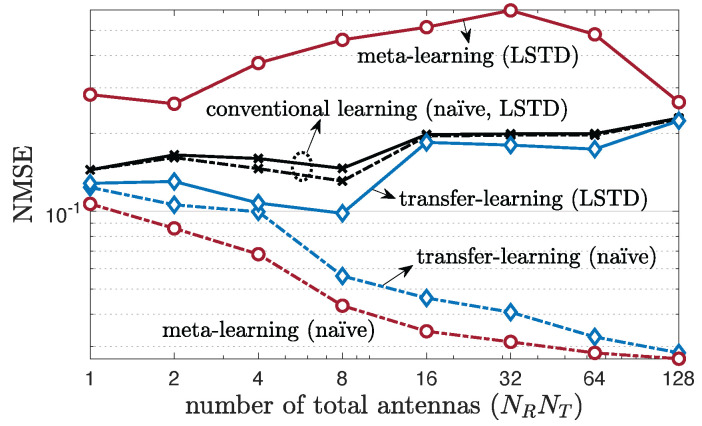
Multi-antenna frequency-flat channel prediction performance as a function of the total number of antennas, $N_{R}N_{T}$, under a single-clustered, single-tap (W=1), 3GPP SCM channel model for a slow-varying environment with number of training samples $L^{\mathrm{new}}=1$ (K=1).

**Figure 6 entropy-24-01363-f006:**
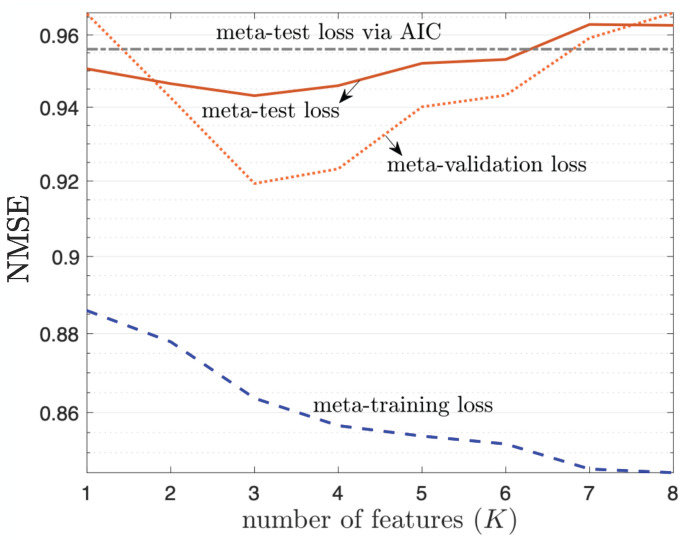
Multi-antenna frequency-selective channel prediction performance as a function of the number of features *K*, under 19-clustered, multi-taps (W=4), multi-antenna ($N_{T}=8,N_{R}=8$) 3GPP SCM channel model for a fast-varying environment with number of training samples $L^{\mathrm{new}}=1$. Results are evaluated with number of previous frames $F^{\mathrm{tr}}=20$ for meta-training, $F^{\mathrm{val}}=20$ for meta-validation, and $F^{\mathrm{te}}=200$ for meta-test.

**Figure 7 entropy-24-01363-f007:**
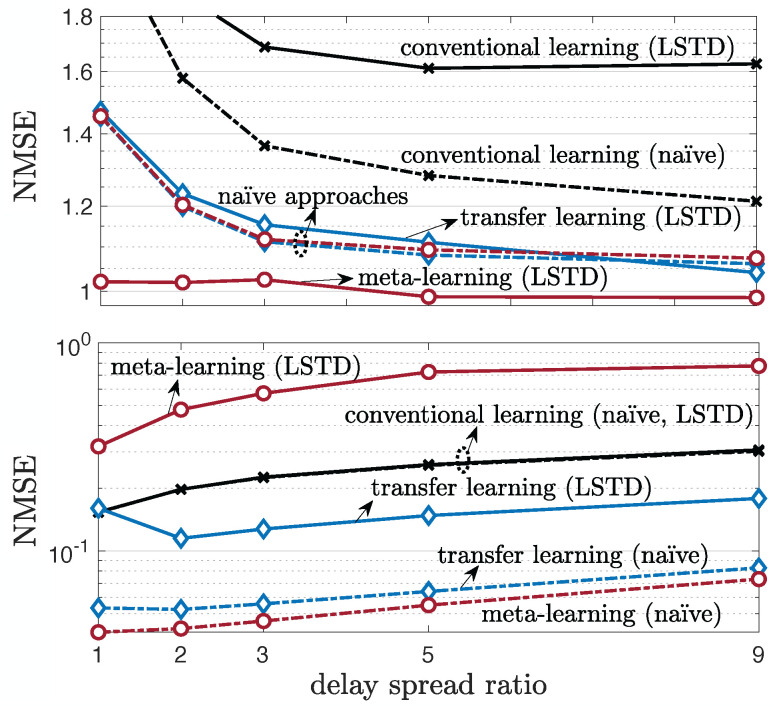
Single-antenna frequency-selective channel prediction performance as a function of delay spread ratio, under 19-clustered, multi-taps, single-antenna ($N_{T}=1,N_{R}=1$) 3GPP SCM channel model for a fast-varying environment (**top**) and slow-varying environment (**bottom**) with number of training samples $L^{\mathrm{new}}=1$ (K=1).

**Figure 8 entropy-24-01363-f008:**
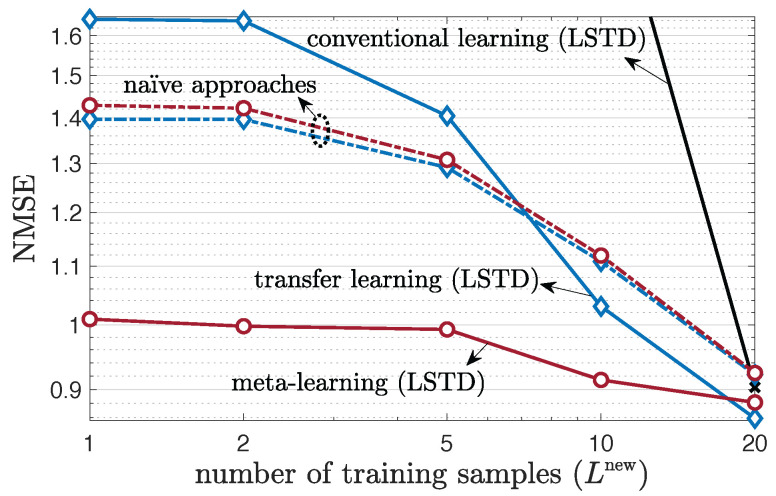
Multi-antenna frequency-selective channel prediction performance as a function of the number of training samples $L^{\mathrm{new}}$, under 19-clustered, two taps (W=2), multi-antenna ($N_{T}=4$, $N_{R}=2$) 3GPP SCM channel model for a fast-varying environment with total number of features K=2 unless determined by Section 5.2.

**Figure 9 entropy-24-01363-f009:**
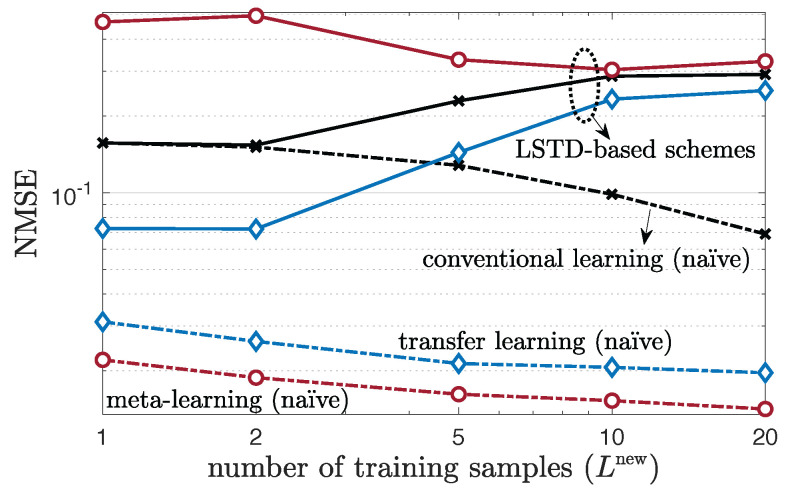
Multi-antenna frequency-selective channel prediction performance as a function of the number of training samples $L^{\mathrm{new}}$, under 19-clustered, two taps (W=2), multi-antenna ($N_{T}=4$, $N_{R}=2$) 3GPP SCM channel model for a slow-varying environment.

**Figure 10 entropy-24-01363-f010:**
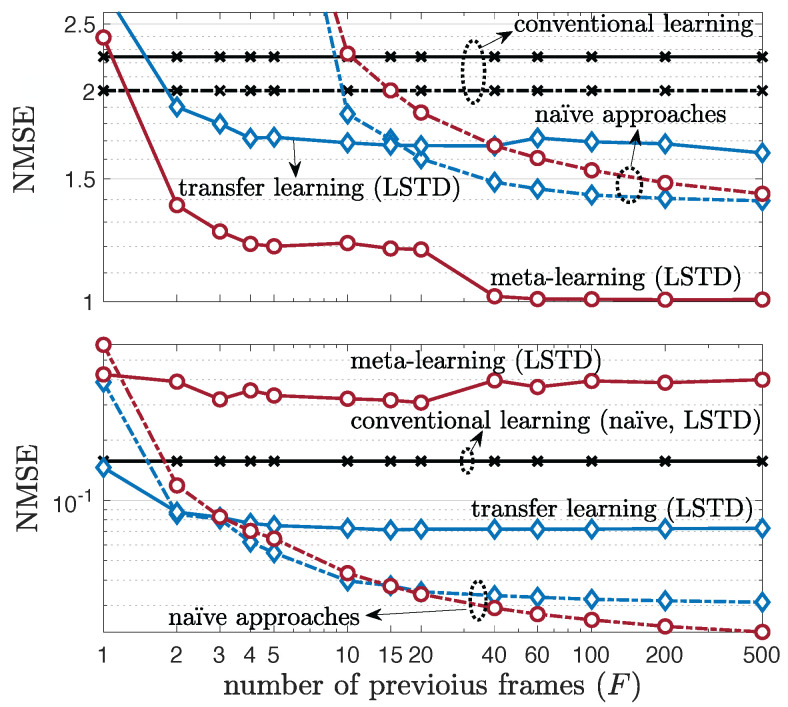
Multi-antenna frequency-selective channel prediction performance as a function of the number of available previous frames *F* under 19-clustered, two taps (W=2), multi-antenna ($N_{T}=4,N_{R}=2$) 3GPP SCM channel model for a fast-varying environment (top) and slow-varying environment (bottom) with number of training samples $L^{\mathrm{new}}=1$.

**Table 1 entropy-24-01363-t001:** Computational complexity analysis at deployment (meta-testing).

Learning Type	O(·) for Naïve Approach	O(·) for LSTD-Based Approach
Conventional learning	$O(S^{3}N^{3}+S^{2}N^{2}L^{\mathrm{tr}})$	$O(KI_{\mathrm{ALS}}\left(L^{\mathrm{tr}}\left(SN^{2}+S^{2}\right)+N^{3}+S^{3}\right))$
Transfer learning	$O(S^{3}N^{3}+S^{2}N^{2}L^{\mathrm{tr}})$	$O(KI_{\mathrm{ALS}}\left(L^{\mathrm{tr}}\left(SN^{2}+S^{2}\right)+N^{3}+S^{3}\right))$
Meta-learning	$O(S^{3}N^{3}+S^{2}N^{2}L^{\mathrm{tr}})$	$O(KI_{\mathrm{ALS}}\left(L^{\mathrm{tr}}\left(SN^{2}+S^{2}\right)+N^{3}+S^{3}\right))$

**Table 2 entropy-24-01363-t002:** Computational complexity analysis during meta-training.

Learning Type	O(·) for Naïve Approach	O(·) for LSTD-Based Approach
Conventional learning	−	−
Transfer learning	$O(S^{3}N^{3}+S^{2}N^{2}FL)$	$O(KI_{\mathrm{ALS}}\left(FL\left(SN^{2}+S^{2}\right)+N^{3}+S^{3}\right))$
Meta-learning	$O(F(S^{3}N^{3}+S^{2}N^{2}L$	$O(KFI_{\mathrm{EP}}I_{\mathrm{ALS}}\left(L\left(SN^{2}+S^{2}\right)+N^{3}+S^{3}\right))$
	$+S^{2}NL^{\mathrm{te}}\left(L^{\mathrm{tr}}+SN\right)))$	

**Table 3 entropy-24-01363-t003:** Experimental setting.

Window size (N)	5
Lag size (δ)	3
Number of previous frames (F)	500
Number of slots (L+N−δ+1)	107
Frequency of the pilot signals ($w_{\mathrm{SRS}}/2\pi$)	200
Normalized Doppler frequency	
for slow-varying environment	ρ∼Unif[0.005,0.05]
Normalized Doppler frequency	
for fast-varying environment	ρ∼Unif[0.1,1]
SNR for channel estimation	20 dB
Number of pilots for channel estimation	100

## Data Availability

Code is available at https://github.com/kclip/channel-prediction-meta-learning (accessed on 22 September 2022).

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
