# Peer review of "Speeding up Training of Linear Predictors for Multi-Antenna Frequency-Selective Channels via Meta-Learning"

_entropy, 2022, doi:10.3390/e24101363_

Round 1

Reviewer 1 Report

This paper studies two data-driven LSTD-based channel predictions, respectively with transfer-learning and meta-learning for multi-antenna frequency selective channels. Overall, this paper is well organized and written, and easy to follow. The considered channel estimation problem is crucial for practical implementation. However, there are many unclear descriptions that limit the contribution of this paper. 

Some comments are given as follows. 

1. The channel estimations in general communication systems have been developed for several decades. In light of the proposed methods, their design concept is similar to the field of adaptive filter design. Therefore, some comments about similar and different parts should be discussed. Also, the proposed method should be compared with such as recursive least square (RLS) and the Kalman channel estimators in terms of both the complexity and the MSE performance. Current simulations are not sufficient to show your superiority over the existing methods. 

2. Also, the proposed methods can also be compared with the model-based channel estimation, such as the expectation-maximization sparse Baysian channel estimation and approximate message passing approaches under the same channel overhead. 

3. Please address the issue of computational complexity for the proposed methods and the existing methods. 

4. Symbols $y_{l,f}$ in (9) present the desired channel, while that in (10) represents the received signals, which confuses the reader. Please modify them accordingly. 

5. $Lambda_1$ and $Lambda_1$ in Algorithms 2 and 3 play as crucial parameters in the design of linear prediction. How to decide it is not mentioned. Also, the parameters are not shown in the Simulation section. Also, how to choose proper initial values in Algorithms 2 and 3? 

Reviewer 2 Report

This paper dealt with channel prediction using data-driven approach to reduce pilot requirements by integrating transfer and meta learning with long-short-term decomposition (LSTD) of linear predictors. About this manuscript, we have two major questions and one minor mistake in formula expression. The manuscript is good to be accepted if the following points can be clarified.

1.  This article is about accelerating multi-antenna antenna estimation through meta learning. However, we do not find the detail about the meta learning. That is, Section 3.3 meta-learning is not enough. The meta-learning descried in this section is too far from the most of actual meta learning.

2.  The manuscript shares many similarity with the paper below. That is, the approach used and application scenario are very similar. We recommend that authors clarify this point in their paper and further explain the contribution of this paper.

"Predicting Multi-Antenna Frequency-Selective Channels via Meta-Learned Linear Filters based on Long-Short Term Channel Decomposition"(https://arxiv.org/abs/2203.12715)

3. According to the definition of H_l,f in (1), Z in (3) is not consistent with the definition of H_l,f. That is, if l=N, Z only can be constructed with h_1, but not the set of {h_1, h_2, ...} as the authors mentioned in the paragraph below (3).

Reviewer 3 Report

This piece of work demonstrated an efficient algorithm of channel prediction with integrating transfer and meta-learning methods. I find the results well-supported by the data and the derivation is presented in a comprehensive manner. 

I recommend publishing this paper in its current form.

Round 2

Reviewer 1 Report

The authors have addressed most of my concerns. However, the computational complexity is not well investigated. The reviewer suggests that the authors can create a subsection and investigate the computational complexity in terms of big-O. The current version regarding the complexity does not provide improvement compared with the previous manuscript. 
